# Aptamer Trimode Biosensor for Trace Glyphosate Based on FeMOF Catalytic Oxidation of Tetramethylbenzidine

**DOI:** 10.3390/bios12110920

**Published:** 2022-10-25

**Authors:** Yuxiang Zhao, Qianmiao Chen, Chi Zhang, Chongning Li, Zhiliang Jiang, Aihui Liang

**Affiliations:** 1School of Public Health, Guilin Medical University, Guilin 541199, China; 2Guangxi Key Laboratory of Environmental Pollution Control Theory and Technology, Guilin 541006, China

**Keywords:** FeMOF nanocatalysis, glyphosate, aptamer, trimode indicator reaction, SERS

## Abstract

The stable and highly catalytic Fe metal–organic framework (FeMOF) nanosol was prepared and characterized by electron microscopy, and energy and molecular spectral analysis. It was found that FeMOF strongly catalyzed the oxidation of 3,3’,5,5’-tetramethylbenzidine (TMB) by H_2_O_2_ to produce TMBox, which had a fluorescence (FL) peak at 410 nm. When silver nanoparticles were added, it exhibited strong resonance Rayleigh scattering (RRS) activity and surface-enhanced Raman scattering (SERS) effect. This new FeMOF nanocatalytic trimode indicator reaction was combined with the glyphosate aptamer reaction to establish a new SERS/RRS/FL trimode biosensor for glyphosate. The sensor can be used for the analysis of environmental wastewater, and a new method for detecting glyphosate content in wastewater is proposed. The linear range of the sensor is 0.1–14 nmol/L, the detection limit is 0.05 nmol/L, the recovery is 92.1–97.5%, and the relative standard deviation is 3.6–8.7%.

## 1. Introduction

Metal–organic frameworks (MOFs) are crystalline porous materials with periodic network structures formed by the self-assembly of transition metal ions and organic ligands. It has the advantages of high porosity, low density, large specific surface area, regular pore channels, adjustable pore size, diversity, and tailorability of the topological structure, and is widely used in gas storage, pharmacology, chemical sensing, pollutant separation and chemical catalysis [1,2,3,4]. Therefore, the preparation and application of MOFs have attracted much attention. Manuel et al. prepared Zn-, Mg-, Mn-, Ni-, and Co-modified MOFs by the aqueous phase method at room temperature using terephthalic acid as a precursor. The results show that the properties of MOFs can be enhanced by reducing the crystal size to nanometer level [5]. Tan et al. proved that copper-based MOF (HKUST-1) exhibits peroxidase activity, using thiamine (TH) as the peroxidase substrate. Based on the peroxidase-like activity of HKUST-1, a simple and sensitive FL detection method for TH was developed. The detection limit was as low as 1 μmol/L and the linear range was 4–700 μmol/L [6]. Zhang et al. synthesized ZIF-8 graphene oxide (ZIF-8 GO) via a simple wet chemical process, and then fixed it onto AuPtNPs by the reduction method to prepare AuPt/ZIF-8 rGO. The peroxidase activity of AuPt/ZIF-8 rGO was used to electrochemically detect 100 nmol/L^−18^ mmol/L H_2_O_2_ with a detection limit (DL) of 19 nmol/L [7]. Wang designed a biomolecular sensing platform based on the peroxidase catalytic activity of MOFs, and established a method for the detection of thrombin according to the significant colorimetric changes, with the DL as low as 0.8 nmol/L. Iron is a common and stable MOF material [8]. Patricia et al. successfully prepared MIL-100 (Fe) by the hydrothermal method. It was proven that the catalytic performance of MIL-100 (Fe) is attributed to the redox characteristics of Fe^2+^ and Fe^3+^, which is similar to the properties of many iron-containing solid catalysts [9]. Zhang et al. prepared Fe-doped metal–organic framework material MIL-100 (Fe) by hydrothermal method, using terephthalic acid as a precursor, and further explored the influence of metal inorganic salt embedding on its catalytic performance. The results showed that the strength of its performance is significantly improved with the addition of metal salts [10]. The Fe-MIL-88NH_2_ prepared by Liu exhibits peroxidase activity, and has been used for the colorimetric determination of glucose. The method is simple and selective, with a linear range of 2 × 10^−6^–3 × 10^−4^ mol/L and a detection limit of 4.5 × 10^−7^ mol/L [11]. Based on FeMOF catalysis of 3, 3′, 5, 5′-tetramethylbenzidine (TMB)–H_2_O_2_ to produce FL, a glucose oxidase coupled with FeMOF, the nanoenzyme FL method was proposed for the assay of glucose [12]. As far as we know, there are no reports regarding the glyphosate aptamer (Apt) reaction, combined with the SERS/RRS/FL trimode indicator reaction of FeMOF–TMB–H_2_O_2_, for the detection of glyphosate.

Apt can bind to a variety of target substances with high specificity and selectivity; thus, it is widely used in the field of biosensors [13,14,15,16]. Recently, the combination of the highly selective Apt reaction and highly sensitive nanocatalytic amplification reaction was discovered to be an important way to construct highly selective and sensitive Apt methods. The detection techniques used include spectrophotometry, FL, RRS, and SERS. Using the catalytic effect of AgCOF and the regulatory effect of Apt on AgCOF, Pan et al. established a simple and sensitive RRS method for the determination of trace melamine based on the RRS signal of its reaction product Cu_2_O. The linear range was 0.79–13.2 nmol/L, and the detection limit was as low as 0.72 nmol/L [15]. Based on the catalytic effect of carbon dots, CDNAg, on the reaction of 3,3′-dimethyl-4,4′-diaminobiphenyl (DBD)–H_2_O_2_ and the regulation of Apt, Feng et al. constructed an analytical platform for SERS detection of acetamiprid (ACT). The linear range was 0.01–1.5 μg/L and the detection limit was 0.006 μg/L [17]. Compared with the single-mode method, the multimode method can provide a variety of detection methods, and has the advantages of the single-mode method while overcoming its disadvantages. Li constructed a DNA enzyme with peroxidase-like activity using Apt. Combined with the molecular reaction of H_2_O_2_–3,3′,5,5′-tetramethylbenzidine (TMB), a SERS/FL dual-mode detection method for K^+^ was constructed. The linear ranges were 2–1000 nmol/L and 10–300 nmol/L, respectively. The detection limits were as low as 1.6 nmol/L and 9.4 nmol/L, respectively [18].

Glyphosate (GLY) is a non-selective and non-residue herbicide, which is very effective for perennial root weeds and is widely used in rubber, mulberry, tea, orchard, and sugarcane fields [19]. It mainly inhibits the enolacetone shikinin phosphate synthase in plants, thereby inhibiting the transformation of shikinin to phenylalanine, tyrosine, and tryptophan, interfering with protein synthesis and leading to plant death [20]. As one of the organophosphate herbicides, it is also associated with neurological diseases and physiological disorders [21]. Therefore, it is of great significance to establish a rapid, sensitive, and simple method for the determination of GLY content [22,23,24]. The commonly used methods for the determination of glyphosate include spectrophotometry, high-performance liquid chromatography (HPLC), HPLC–mass spectrometry, ion chromatography, gas chromatography, gas chromatography–mass spectrometry, capillary electrophoresis, and enzyme-linked immunosorbent assay. With the development of science and technology, some new methods have also been reported. Hou et al. established a FL detection method for glyphosate based on the dynamic quenching of FL nanoprobes by carbon dots, combined with the complexation of functional groups in GLY molecules with Fe^3+^, and the regulation of aptamers. The linear range was 0.1–16 ppm, and the DL was as low as 8.75 ppb [25]. Songa et al. electrochemically deposited poly(2,5-dimethoxyaniline) (PDMA) doped with 4-styrene sulfonic acid (PSS) on the surface of gold electrode, and then electrostatically adsorbed horseradish peroxidase (HRP) on the PDMA–PSS composite membrane. With the help of the inhibitory effect of glyphosate on horseradish peroxidase, a biosensor for the determination of GLY was prepared. The determination range was 0.25–14 μg/L, and the DL was as low as 1.7 μg/L [26]. Some of these detection methods have low sensitivity and some have a long analysis process [27]. The trimode method not only overcomes some of the disadvantages of the single-mode method while retaining its advantages, but also provides three methods for selection. As far as we know, there is no report on the synthesis of FeMOF with high catalytic activity, or the nanocatalytic amplification of the H_2_O_2_–TMB indicator reaction and its coupling with highly selective aptamers to establish trimode detection. In this study, a novel trimode SERS method for the determination of trace GLY was developed by combining the highly selective aptamer reaction with the sensitive FeMOF nanocatalytic H_2_O_2_–TMB trimode indicator reaction. It has the advantages of convenient operation, good selectivity, high sensitivity, and can provide trimodal options. The SERS method was applied to the determination of GLY in wastewater with satisfactory results.

## 2. Materials and Methods

### 2.1. Apparatus

The resonance Rayleigh scattering spectrum was scanned by Hitachi F-7000 fluorescence spectrophotometer (Hitachi Hi-tech Company, Hitachi, Japan), and the absorption spectrum was scanned by Tu-1901 dual-beam ultraviolet–visible spectrophotometer (Beijing Pusan General Instrument Co., Ltd., Beijing, China). The DXR smart Raman spectrometer (Thermo, Waltham, MA, USA) has a laser wavelength of 633 nm, a laser power of 0.5 mw, a slit of 2.5 μm, and an intelligent background scanning mode to obtain SERS spectra, which includes a scanning electron microscope (SEM, JSM-6380LV, Hitachi, Japan, Chiyoda, Tokyo, Japan) and a transmission electron microscope (FEI Talos 200S, Thermo, Waltham, MA, USA). An FD-1C-50 vacuum freeze dryer (Hangzhou Jutong Electronics Co., Ltd., Hangzhou, China) was used to prepare solid samples, and the infrared spectra were obtained by scanning with a Fourier transform infrared spectrometer (Shanghai Platinum Elmer Co., Ltd., Shanghai, China). The size spectra of nanomaterials were obtained by Nano-2s nanoparticles (Malverm Co., Malverm, UK). Other instruments include an HH-S2 electrothermal constant-temperature water bath (Jintan Dadi Automation Instrument Factory, Jintan, China), used for heating the reaction system, and a KQ3200DB numerical control ultrasonic cleaner with ultrasonic power of 150 w and operating frequency of 40 KHz (Kunshan Ultrasonic Instrument Co., Ltd., Kunshan, China), used for cleaning the required equipment for the experiment.

### 2.2. Reagents

AgNO_3_ (Guangzhou Guanghua Technology Co., Ltd., Guangzhou, China); trisodium citrate (Shanghai Reagent Co., Ltd., Shanghai, China); NaBH_4_ (Shanghai Chemical Reagent Co., Ltd., Shanghai, China); 3,3′,5,5′-tetramethylbenzidine (TMB, Shanghai McLin Biochemical Technology Co., Ltd., Shanghai, China, C_16_H_20_N_2_); hydrogen peroxide (H_2_O_2_, Xilong Science Co., Ltd., Shantou, China); trimethylaminomethane (Tris, Shanghai Yuanye Biotechnology Co., Ltd., Shanghai, China); hydrochloric acid (Sichuan Xilong Science Co., Ltd., Sichuan, China); ferrous sulfate heptahydrate (FeSO_4_·7H_2_O, Xilong Chemical Co., Ltd., Shantou, China); 1,3,5-benzenetricarboxylic acid (H_3_btc, Shanghai McLin Biochemical Technology Co., Ltd., Shanghai, China); sodium hydroxide (NaOH, Xilong Chemical Co., Ltd., Shantou, China); anhydrous ethanol (Chengdu Colon Chemicals Co., Ltd., Chengdu, China); glyphosate (GLY, Beijing Bailingwei Technology Co., Ltd., Beijing, China, C_3_H_8_NO_5_P, relative molecular mass: 169.07). The sequence number of the glyphosate aptamer (Apt_GLY_) is TGC TAG ACG ATA TTC CAT CCG AGC CCG TGG CGG GCT TTA GGA CTC TGC GGG CTT CGC GGC GCT GTC AGA CTG ATG TCA (Shanghai Bioengineering Co., Ltd., Shanghai, China). The above reagents were analytically pure, and the experimental water was deionized water.

### 2.3. Preparation of Silver Nanoparticles

Into a conical flask, 44 mL water, 2 mL 1.7 g/L AgNO_3_, 2.0 mL 100 mmol/L sodium citrate, 600 μL 30% H_2_O_2_, and 600 μL 0.1 mol/L NaBH_4_ were added in sequence while stirring, the black nanosilver colloid was quickly stirred to obtain a black color. The black nanosilver colloid was immediately transferred to the light wave furnace, and the orange-red transparent nanosilver colloid was obtained by 250 °C light wave for 10 min. After natural cooling, the water was fixed to 50 mL for standby. The concentration was 68 mg/L AgNPs. The reagents used were all analytical pure, and the experimental water was sub-boiling water.

### 2.4. Preparation of FeMOF/CoMOF/CuMOF/NiMOF

First, 0.53 g NaOH was added into the round-bottom flask and dissolved in 13 mL water, after which 0.735 g H_3_btc was added to the flask in batches and dissolved by ultrasound. Subsequently, 1.5 g FeSO_4_·7H_2_O was weighed and dissolved in a 100 mL round-bottom flask with 27 mL H_2_O ultrasonically. Under ultrasonic stirring, FeSO_4_ solution was slowly added to the H_3_btc solution at room temperature, and continuously stirred for 12 h to obtain a red-brown suspension. The suspension was centrifuged at 12,000 r/min to obtain red-brown products, and then the products were washed three times with deionized water and anhydrous ethanol, respectively. Then, the red-brown solid was obtained by high-speed freezing centrifugation (12,000 r/min). Finally, the product was vacuum freeze-dried for 24 h to obtain red-brown powder, namely FeMOF. The preparation of CoMOF/CuMOF/NiMOF followed the same steps as that of FeMOF, except replacing 1.5 g FeSO_4_·7H_2_O with CoCl_2_·6H_2_O, CuSO_4_·H_2_O, and NiCl_2_·6H_2_O of the same mass. A 0.01 g/L FeMOF solution was prepared with water.

### 2.5. Experimental Procedure

Into a 5.0 mL test tube, 150 μL 0.01 g/L FeMOF, 200 μL 0.1 μmol/L Apt_GLY_, and a certain concentration of glyphosate solution were added. The test tube was then placed in an 80 °C water bath for 5 min to allow the Apt to combine with the catalyst, after which was then added 100 μL 0.5 mmol/L TMB solution, 90 μL 0.1 mmol/L H_2_O_2_ solution, and 90 μL 0.1 mol/L Tris-HCl buffer. Subsequently, the solution was diluted to 2.0 mL with water, and mixed. After bathing at 45 °C for 30 min, the reaction was terminated in ice water, and 400 μL 68 mg/L AgNPs were added. The solution was placed into a quartz cell, and the Raman spectrum was recorded under the condition of 1 mW light source power and a 25.0 μm slit. The SERS intensity at 1607 cm^−1^ was measured, and the blank (I_1607_ cm^−1^) _0_ was not mixed with GLY. The value of ΔI_1607_ cm^−1^ = I_1607_ cm^−1^ − (I_1607_ cm^−1^) _0_ was calculated.

## 3. Results and Discussion

### 3.1. Analysis Principles

FeMOF can catalyze the oxidation of TMB by H_2_O_2_ to produce TMBox. TMBox can make AgNPs aggregate, and its SERS activity and RRS effect are greatly enhanced. When Apt is added, it can be adsorbed onto the surface of FeMOF to inhibit its catalytic effect. The TMBox generated in the system decreases, and the SERS/RRS/FL signals decrease. When the target molecule GLY is added, it forms a stable complex with Apt and leaves the nanosurface. Its catalytic effect is enhanced, and the TMBox generated by the system is increased, and the SERS/RRS/FL signal is linearly enhanced. Accordingly, a new Apt-mediated FeMOF catalytic amplification RRS/SERS/FL trimode quantitative detection method for GLY can be established using the Raman spectrometer (Raman) and fluorescence instrument (FM) (Figure 1).

### 3.2. Characterization of FeMOF

Figure 2A shows the absorption spectra of FeMOF, indicating that FeMOF has a wide absorption band at 300–650 nm. With the increase in FeMOF concentration, the absorbance gradually increases, and an absorption peak appears at 375 nm. The resonance Rayleigh scattering spectra (RRS, volt = 350 V, excited slit = emission slit = 5 nm) of FeMOF were obtained by fluorescence synchronous scanning technique. It is obvious that the RRS signal at 440 nm increases with the increase in FeMOF concentration, indicating that FeMOF has an RRS signal, which lays a foundation for the establishment of subsequent RRS analysis method (Figure 2B). The prepared FeMOF powder was ground uniformly in an agate bowl and placed into a Petri dish. The FeMOF powder was dried again in the oven at 50 °C for 2 h to remove the moisture that may have been absorbed during the transfer process. The powder was pressed by the tablet press. The tablet was placed into the infrared spectrometer to scan and obtain the corresponding spectrum. The infrared spectra showed that for FeMOF, strong infrared spectral peaks were mainly generated at 1622.1 cm^−1^, 1383.5 cm^−1^, 1109.8 cm^−1^, 1044.9 cm^−1^, 943.6 cm^−1^, 759.6 cm^−1^, and 710.2 cm^−1^, respectively. The broad peak, 3445 cm^−1^, belongs to the hydroxyl -OH peak, the peaks at 1622.1 cm^−1^, 1383.5 cm^−1^, 1109.8 cm^−1^, and 1044.9 cm^−1^ belong to the stretching vibration peak of C=O, the bending vibration peak of hydroxyl, and the stretching vibration peak of C-O, respectively. Peaks at 759.6 cm^−1^ and 710.2 cm^−1^ belong to the fingerprint peak of FeMOF and the structure of Fe^3+^ oxide, respectively. It was proven that FeMOF was successfully prepared. In order to prove that the synthesized material is FeMOF, the crystal structure was analyzed by X-ray powder diffraction (XRD). Figure 2D presents the XRD pattern of FeMOF powder synthesized in aqueous phase at room temperature for 12 h and 24 h. When the visible reaction proceeded, as displayed in Figure 2C, to 12 h, the diffraction peak intensity of the spectrum was high, and the diffraction characteristic peak of FeMOF appeared, especially in the range of 3–5°. The main diffraction peaks corresponded to the peak position of FeMOF reported in literature, indicating that FeMOF was successfully synthesized. When the reaction proceeded to 24 h, the intensity of diffraction peak decreased, indicating that the crystallinity of FeMOF decreased. Therefore, the reaction time of 12 h is the optimal time for the synthesis of FeMOF powder, and the prepared FeMOF has a good crystal form. The particle size distribution of nanoparticles in FeMOF solution was analyzed by Nano-2s particle size analyzer. The laser scattering size distribution of FeMOF solution was recorded by the nanoparticle size and potential analyzer. The size distribution was from 820 nm to 1200 nm, with an average size of 955 nm (Figure 2E), which is consistent with the particle size range of FeMOF obtained by SEM. FeMOF was prepared according to the experimental method, and 10 mg was dissolved in water and ultrasonically dispersed to obtain 10 mL of yellow suspension. After 10-fold dilution, 5 μL solution was placed on the silicon wafer and recorded by scanning electron microscopy (SEM). From Figure 2F, it can be seen that FeMOF tends to be a cubic sheet structure, and the average particle size is between 500 nm^−1^ μm.

### 3.3. SERS Spectra

In pH 4.0 Tris-HCl buffer solution, AuNPs, AgNPs, HPR, CoMOF, CuMOF, NiMO, and FeMOF can catalyze the oxidation of TMB by H_2_O_2_ at 45 °C in a water bath. The catalytic product, TMBox, has a SERS effect (Figure 3). The results show that the catalyst concentration was linear to the SERS peak intensity at 1607 cm^−1^ (Appendix A). The catalytic effect was best when FeMOF was used as the catalyst, thus FeMOF was selected as the catalyst. After adding Apt, FeMOF can be wrapped, which inhibits the catalytic ability of FeMOF, reduces the generated TMBox, and weakens the SERS intensity. When the target molecule, GLY, was added, GLY specifically bound with Apt and released FeMOF, and its catalytic activity was restored. Ag nanoparticle has the properties of surface effect, volume effect, quantum size effect, and the macroquantum tunneling effect of general metal nanomaterials, and also has the special effect of surface plasmon resonance. When the Ag nanoparticle was irradiated by the laser, the very small scale of Ag nanoparticle makes the nanosphere cavity array cooperate with surface plasmon resonance and optical coupling, resulting in the enhancement of electromagnetic field. Therefore, the Raman signal of adsorbed molecules on the Ag nanosurface is 10^6^ times, or even stronger, than that of normal molecules, which makes it one of the best SERS metal substrates. When a certain concentration of photo-silver nanoparticles was added, the system showed strong Raman peaks at 1184 cm^−1^, 1332 cm^−1^, and 1607 cm^−1^, respectively. Among them, the SERS peak at 1607 cm^−1^ had the most obvious change and good linearity. Therefore, the SERS peak at 1607 cm^−1^ was selected for the detection of glyphosate, and the signal intensity was linear with the added amount of GLY.

### 3.4. RRS and Fluorescence Spectra

RRS and fluorescence spectra were obtained by fluorescence spectrophotometer. The former adopted synchronous scanning technology, and the latter fixed excitation wavelength to scan emission wavelength. In pH 4.0 Tris-HCl buffer solution, FeMOF can catalyze the oxidation of TMB by H_2_O_2_ to generate TMBox in a 45 °C water bath. The oxidation product TMBox can make silver nanoparticles aggregate. With the increase in TMBox, the RRS signal of the system gradually increases (Figure 4A and Appendix AA). After the addition of Apt to GLY, Apt wrapped FeMOF, thereby inhibiting the catalytic ability of FeMOF (Figure 4B). The generated TMBox decreased, thus the RRS strength decreased accordingly. For the H_2_O_2_–TMB–Tris-HCl reaction system, the target GLY specifically binds to Apt, FeMOF is released, and the catalytic activity is gradually restored. The signal intensity of RRS spectrum is linear with the amount of GLY (Figure 4C). The fluorescence spectra in Figure 4D show that FeMOF had a strong catalytic effect on the reduction of TMB by H_2_O_2_. Moreover, with the increase in FeMOF concentration, the signal value is stronger and has an obvious linear relationship (Appendix A). Apt_GLY_ has a strong inhibitory effect on the nanocatalytic system; with the increase in Apt_GLY_ concentration, the stronger the inhibition. Figure 4F shows that the aptamer-mediated FeMOF catalysis fluorescence can determine glyphosate. Furthermore, with the increase in GLY concentration, the signal value is stronger, and therefore has an obvious linear relationship.

### 3.5. Transmission Electron Microscopy (TEM) and Laser Scattering of the Analytical System

The system reaction liquid was obtained according to the following test method: place the liquid into a 2 mL centrifuge tube, centrifugate at 10,000 rpm/g for 5 min; take the supernatant, add 1.5 mL deionized water, and centrifugate twice; the precipitation of 1.5 mL deionized water occurs via ultrasonic dispersion; take 5 μL solution and drop onto the silicon wafer, commence scanning electron microscopy. It can be seen from the TEM images that AgNPs are relatively dispersed with an average particle size of about 30 nm (Figure 5A). For the analysis system, when glyphosate was not detected, AgNPs continued to exist in the form of dispersion (Figure 5B), and the particle size was about 30 nm. When glyphosate was added into the reaction system, more TMBox was generated, which made AgNPs aggregate into larger particles (Figure 5C), with a particle size of about 50 nm. The Nano-2s particle size analyzer was used to determine the particle size distribution of the nanoparticles in the system. After adding glyphosate, glyphosate formed a stable complex with Apt_GLY_ and released encapsulated FeMOF particles. The catalytic activity of FeMOF was restored, and the TMBox generated in the system was gradually increased, and the aggregation of TMBox-AgNPs was enhanced. The particle sizes of the H_2_O_2_–TMB–Tris-HCl–FeMOF–Apt_GLY_ system and the H_2_O_2_–TMB–Tris-HCl–FeMOF–Apt_GLY_–GLY system were 38 nm and 460 nm, respectively (Figure 5D), and the particle sizes showed an increasing trend. The two particle sizes of glyphosate system are quite different; this is because TEM is a solid-state imaging method, laser scattering measurement is the average value of the solution state.

### 3.6. Nanocatalytic Mechanism of FeMOF

Generally speaking, the smaller the particle size of nanoparticles, the stronger the catalysis. The results of the electron microscopy and laser scattering particle size distribution experiments show that FeMOF has a large particle size, but it still has a strong catalytic effect on the oxidation of TMB to TMBox by H_2_O_2_. This is related to the porous structure and nanopore size of FeMOF. That is, its nanopores can provide more nanosurface electrons. It is usually difficult to react when H_2_O_2_ oxidizes TMB. When FeMOF is added, H_2_O_2_ will react with TMB, and with the increase in nanocatalyst concentration, the catalytic capacity will increase, and the amount of TMBox generated by catalysis will increase. After adding a certain concentration of silver nanoparticles (AgNPs) sol, TMBox can make AgNPs aggregate, and enhance its SERS activity and RRS effect. The large specific surface area of FeMOF provides a large number of reaction sites for the reaction between H_2_O_2_ and TMB. The porous structure of FeMOF can adsorb the reactants to the nanosurface of the nanomaterial, the electrons on the nanopores speed up the redox electron transfer to enhance the H_2_O_2_–TMB reaction. At the same time, Fe(II) on the catalyst surface will activate hydrogen peroxide to generate ·OH radicals and Fe(III) [28,29], Fe(III)MOF will further react with H_2_O_2_ to generate HO_2_· and Fe(II)MOF, and the circulation of Fe(II)MOF and Fe(III)MOF on the catalyst surface plays the role of a nanocatalyst (Figure 6).
(1)$$Fe\left(II\right)MOF+H_{2}O_{2}\to \;Fe\left(III\right)MOF+OH+OH^{-}$$
(2)$$Fe\left(III\right)MOF+H_{2}O_{2}\;\to \;Fe\left(II\right)MOF+HO_{2}+H^{+}$$
(3)$$Fe\left(III\right)MOF+TMB\;\to \;Fe\left(II\right)MOF+TMB_{OX}$$
(4)$$OH+TMB\;\to \;TMB_{OX}$$
(5)$$HO_{2}+TMB\;\to \;TMB_{OX}$$

### 3.7. Selection of the Preparation and Analytical Conditions

#### 3.7.1. Preparation Conditions of FeMOF

The preparation conditions of FeMOF were selected by single-factor transformation. The three factors of trimellitic acid, sodium hydroxide, and ferrous sulfate hexahydrate were investigated (Appendix A). Based on the I_1607_ cm^−1^ of the H_2_O_2_–TMB system, FeMOF was prepared by heating 0.398 g, 0.585 g, 0.735 g, and 0.815 g trimellitic acid to room temperature, and stirring for 12 h according to the method. The effect of trimellitic acid addition on the catalytic effect of H_2_O_2_–TMB system was investigated. With the increase in trimellitic acid content, SERS signal value first increased and then decreased. When the addition amount of trimellitic acid was 0.735 g, I_1607_ cm^−1^ was the largest. Therefore, the amount of trimellitic acid added was 0.735 g. The effects of 0.2 g, 0.36 g, 0.53 g, and 0.84 g sodium hydroxide on the catalytic effect of H_2_O_2_–TMB system were investigated. With the increase in sodium hydroxide content, SERS signal value first increased and then decreased. When the addition amount of sodium hydroxide was 0.53 g, I_1607_ cm^−1^ was the largest. Therefore, the addition amount of sodium hydroxide was 0.53 g. The effects of 0.6 g, 1.2 g, 1.5 g, and 2.1 g ferrous sulfate heptahydrate on the catalytic effect of H_2_O_2_–TMB system were investigated. With the increase in ferrous sulfate heptahydrate content, the SERS signal value first increased and then decreased. When the addition amount of ferrous sulfate heptahydrate was 1.5 g, I_1607_ cm^−1^ was the largest. Therefore, the addition amount of ferrous sulfate heptahydrate was 1.5 g.

#### 3.7.2. Analysis Conditions

The analytical conditions were examined. The effect of H_2_O_2_ concentration on the SERS signal of the system was investigated. When 4.5 × 10^−3^ mmol/L H_2_O_2_ was added, ΔI reached the maximum; therefore, 4.5 × 10^−3^ mmol/L H_2_O_2_ was selected (Appendix A). The effect of TMB concentration on the SERS signal of the system was investigated. When 0.025 mmol/L TMB was added, ΔI reached the maximum, this this concentration was chosen (Appendix A). The effect of Tris-HCl buffer solution concentration on the SERS signal of the system was investigated. When 0.45 mmol/L Tris-HCl buffer solution was added, ΔI reached the maximum level, therefore, this concentration is the optimal concentration (Appendix A). The effect of Apt_GLY_ concentration on the SERS signal of the system was investigated. When 8 nmol/L Apt_GLY_ was added, ΔI reached the maximum, hence 8 nmol/L AptGLY was selected (Appendix A). The effect of reaction temperature on the SERS signal of the system was investigated. When the reaction time is 50 °C, ΔI reached the maximum level, thus the optimum temperature is 50 °C (Appendix A). The effect of reaction time on the SERS signal was considered. When the reaction time is 30 min, ΔI reached the maximum level, and thus, 30 min was chosen (Appendix A).

### 3.8. Working Curve

Under the selected experimental conditions, for the H_2_O_2_–TMB–FeMOF–Apt_GLY_–GLY–AgNPs SERS system, in the concentration range of 0.1–14 nmol/L GLY (Figure 7A), the SERS intensity change ΔI_1607_ cm^−1^ was linear to the concentration of GLY, and the linear equation was ΔI_1607_ cm^−1^ = 722C − 49.2. The linear correlation coefficient, R^2^, was 0.9947, and the DL was 0.05 nmol/L. For the H_2_O_2_–TMB–FeMOF–Apt_GLY_–GLY–AgNPs RRS system, in the concentration range of 1–10 nmol/L GLY, the change of RRS intensity ΔI at 370 nm was linear with the concentration of GLY. The linear equation was ΔI_370 nm_ = 82.2C + 18.1, the coefficient R^2^ was 0.9950, and the DL was 0.5 nmol/L (Figure 7B). For the FL analysis system, in the concentration range of 2.0–10 nmol/L GLY, the FL intensity change ΔF at 410 nm was linearly related to the GLY concentration. The linear equation was ΔF_410 nm_ = 68.9C + 12, the coefficient R^2^ was 0.9748, and the DL was 1.0 nmol/L (Figure 7C). With the trimode method, the SERS was most sensitive and the linear range was the widest, the RRS was second, and the equipment cost was lower that the Raman meter. In addition, the FM can be finished the RRS/FL dimode detection. Table 1 lists the reported methods for determination of glyphosate, including the method, detection limit, recovery, and analysis sample. This SERS method was sensitive.

### 3.9. Influence of Coexisting Substances

According to the experimental method, the effect of interfering substances on the determination of 2.5 ng/L glyphosate was detected. The results presented in Appendix A show that when the relative error is less than ±10%, the common inorganic ions and organic compounds have little effect on the measurement results of glyphosate, indicating that the method has good selectivity.

### 3.10. Analysis of Samples

The wastewater sample was filtered by 100 mL, and the supernatant was diluted to obtain the sample solution. According to the experimental method, 100 μL sample solution was taken to determine glyphosate content (Appendix A). The SERS analytical results were in agreement with the HPLC results. The relative standard deviation (RSD) was 3.6–8.7%, and the recovery was 92.1–97.5%. This indicates that this SERS assay was accurate and reliable.

## 4. Conclusions

In this study, FeMOF prepared by aqueous phase method at room temperature exhibited good catalytic performance, which can facilitate TMB oxidization by H_2_O_2_ to produce more TMBox probe molecules. Through the regulation of nucleic acid aptamers, an effective SERS/RRS/FL trimode detection method for GLY in soil and water environments was established. This method combines the characteristics of SERS detection with low DL and good accuracy of RRS detection, and can be applied to the detection of GLY in actual samples. The recovery rate is between 92.1% and 97.5%, and the results are satisfactory. In addition, a reasonable nanocatalytic mechanism was proposed, that the surface electrons on FeMOF nanopores enhanced the redox electron transfer efficiency.

## Figures and Tables

**Figure 1 biosensors-12-00920-f001:**
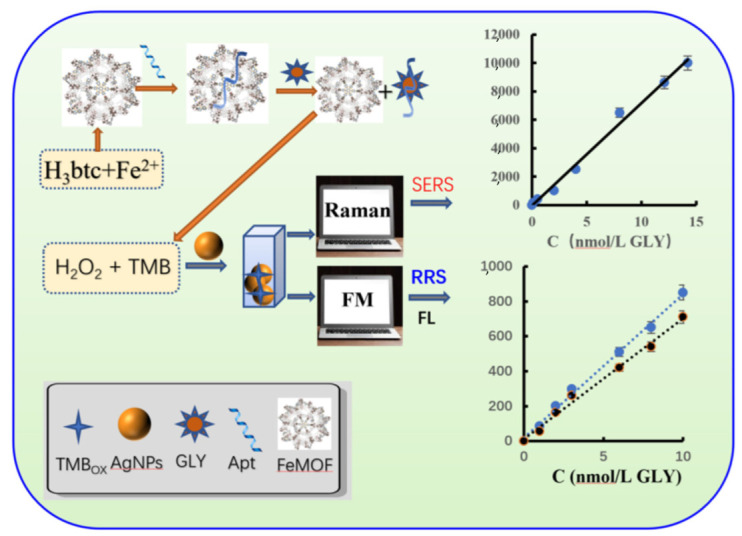
Principle of FeMOF catalytic amplification RRS/SERS/FL trimode detection of GLY.

**Figure 2 biosensors-12-00920-f002:**
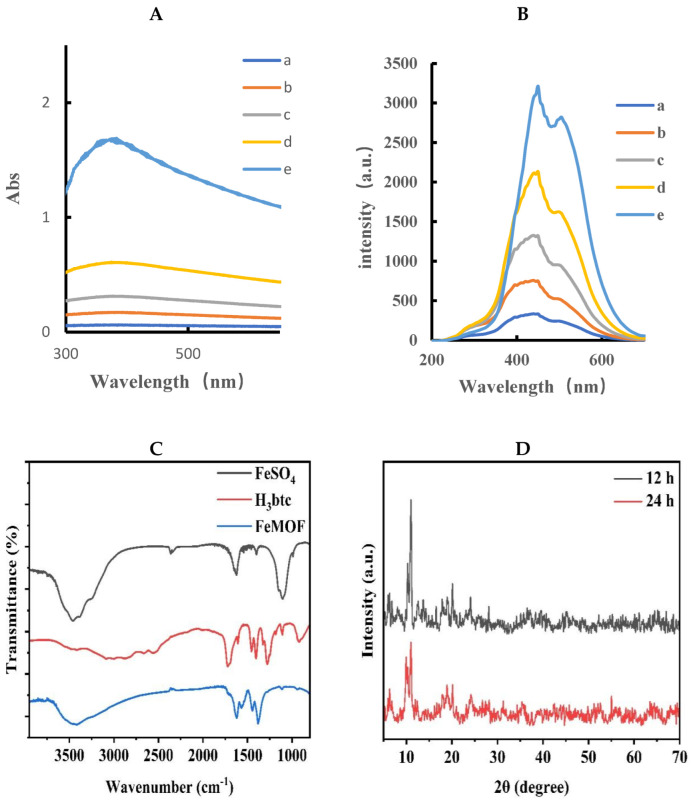
Absorption and RRS spectra of FeMOF. (**A**) Absorption spectrum, a: 0.01 mg/mL FeMOF; b: 0.025 mg/mL FeMOF; c: 0.05 mg/mL FeMOF; d: 0.1 mg/mL FeMOF; e: 0.25 mg/mL FeMOF. (**B**) RRS spectrum, a: 0.01 mg/mL FeMOF; b: 0.025 mg/mL FeMOF; c: 0.05 mg/mL FeMOF; d: 0.1 mg/mL FeMOF; e: 0.25 mg/mL FeMOF. (**C**) Infrared spectrum. (**D**) XRD. (**E**) Particle size distribution. (**F**) Scanning electron microscopy image.

**Figure 3 biosensors-12-00920-f003:**
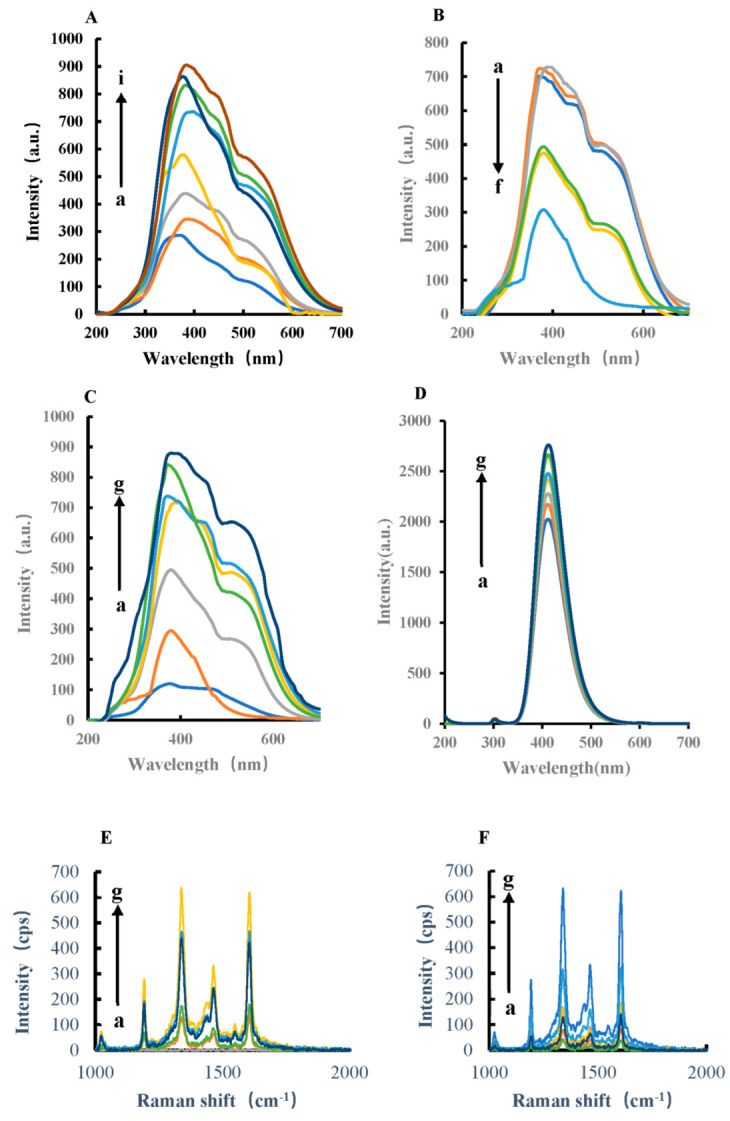
SERS spectra of H_2_O_2_−TMB−AuNPs/AgNPs/HPR/CoMOF/CuMOF/NiMOF/ FeMOF−AgNPs catalytic system and H_2_O_2_−TMB−FeMOF−Apt_GLY_−GLY−AgNPs analytical system. (**A**) a: 0.45 mmol/L Tris-HCl + 4.5 × 10^−6^ mol/L H_2_O_2_ + 0.025 mmol/L TMB + 11.6 mg/L AgNPs; b: a + 0.003 mg/L AuNPs; c: a + 0.015 mg/L AuNPs; d: a + 0.03 mg/L AuNPs; e: a + 0.06 mg/L AuNPs; f: a + 0.15 mg/L AuNPs; g: a + 0.3 mg/L AuNPs. (**B**) a: 0.45 mmol/L Tris-HCl + 4.5 × 10^−6^ mol/L H_2_O_2_ + 0.025 mmol/L TMB + 11.6 mg/L AgNPs; b: a + 1.725 mg/L AgNPs; c: a + 3.5 mg/L AgNPs; d: a + 7.1 mg/L AgNPs; e: a + 18 mg/L AgNPs; f: a + 36 mg/L AgNPs. (**C**) a: 0.45 mmol/L Tris-HCl + 4.5 × 10^−6^ mol/L H_2_O_2_ + 0.025 mmol/L TMB + 11.6 mg/L AgNPs; b: a + 0.025 g/L HPR; c: a + 0.05 mg/L HPR; d: a + 0.1 mg/L HPR; e: a + 0.25 mg/L HPR; f: a + 0.5 mg/L HPR; g: a + 1 mg/L HPR; h: a + 2.5 mg/L HPR. (**D**) a: 0.45 mmol/L Tris-HCl + 4.5 × 10^−6^ mol/L H_2_O_2_ + 0.025 mol/L TMB + 11.6 mg/L AgNPs; b: a + 0.25 mg/L CoMOF; c: a + 0.75 mg/L CoMOF; d: a + 1.5 mg/L CoMOF; e: a + 7.5 mg/L CoMOF; f: a + 0.015 g/L CoMOF; g: a + 0.025 g/L CoMOF. (**E**) a: 0.45 mmol/L Tris-HCl + 4.5 × 10^−6^ mol/L H_2_O_2_ + 0.025 mol/L TMB + 11.6 mg/L AgNPs; b: a + 0.25 mg/L CuMOF; c: a + 0.75 mg/L CuMOF; d: a + 1.5 mg/L CuMOF; e: a + 2.5 mg/L CuMOF; f: a + 7.5 mg/L CuMOF; g: a + 0.015 g/L CuMOF. (**F**) a: 0.45 mmol/L Tris-HCl + 4.5 × 10^−6^ mol/L H_2_O_2_ + 0.025 mol/L TMB + 11.6 mg/L AgNPs; b: a + 0.25 mg/L NiMOF; c: a + 0.75 mg/L NiMOF; d: a + 1.5 mg/L NiMOF; e: a + 2.5 mg/L NiMOF; f: a + 7.5 mg/L NiMOF; g: a + 0.015 g/L NiMOF; (**G**) a: 0.45 mmol/L Tris-HCl + 4.5 × 10^−6^ mol/L H_2_O_2_ + 0.025 mmol/L TMB + 11.6 mg/L AgNPs; b: a + 0.25 mg/L FeMOF; c: a + 0.75 mg/L FeMOF; d: a + 1.5 mg/L FeMOF; e: a + 2.5 mg/L FeMOF; f: a + 7.5 mg/L FeMOF; g: a + 0.015 g/L FeMOF; h: a + 0.025 g/L FeMOF; i: a + 0.05 g/L FeMOF. (**H**) a: 0.45 mmol/L Tris-HCl + 4.5 × 10^−6^ mol/L H_2_O_2_ + 0.025 mmol/L TMB + 7.5 mg/L FeMOF + 11.6 mg/L AgNPs; b: a + 0.05 nmol/L Apt_GLY_; c: a + 0.1 nmol/L Apt_GLY_; d: a + 0.5 nmol/L Apt_GLY_; e: a + 1 nmol/L Apt_GLY_; f: a + 5 nmol/L Apt_GLY_; g: a + 0.01 µmol/L Apt_GLY_; h: a + 0.025 µmol/L Apt_GLY_; i: a + 0.05 µmol/L Apt_GLY_. (**I**) a: 0.45 mmol/L Tris-HCl + 4.5 × 10^−6^ mol/L H_2_O_2_ + 0.025 mmol/L TMB + 7.5 mg/L FeMOF + 0.01 µmol/L Apt_GLY_ + 11.6 mg/L AgNPs; b: a + 0.1 nmol/L GLY; c: a + 0.5 nmol/L GLY; d: a + 2 nmol/L GLY; e: a + 4 nmol/L GLY; f: a + 8 μmol/L GLY; g: a + 10 nmol/L GLY; h: a + 14.22 nmol/L GLY.

**Figure 4 biosensors-12-00920-f004:**
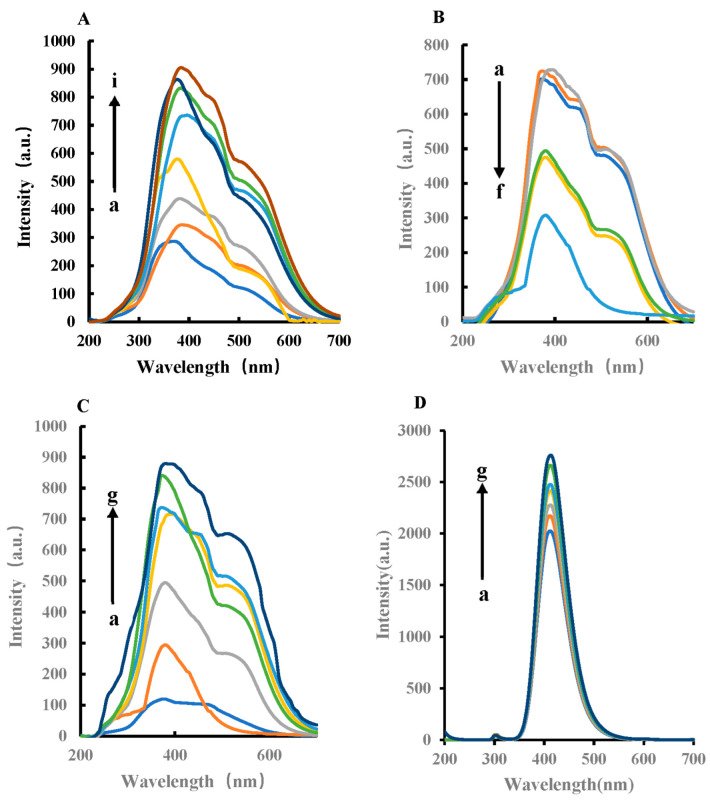
RRS and fluorescence spectra of the H_2_O_2_–TMB–FeMOF–Apt_GLY_–GLY–AgNPs system. (**A**) RRS spectra, a: 0.45 mmol/L Tris-HCl + 4.5 × 10^−6^ mol/L H_2_O_2_ + 0.025 mmol/L TMB + 11.6 mg/L AgNPs; b: a + 0.25 mg/L FeMOF; c: a + 0.75 mg/L FeMOF; d: a + 1.5 mg/L FeMOF; e: a + 2.5 mg/L FeMOF; f: a + 7.5 mg/L FeMOF; g: a + 0.015 g/L FeMOF; h: a + 0.025 g/L FeMOF; i: a + 0.05 g/L FeMOF. (**B**) RRS spectra, a: 0.45 mmol/L Tris-HCl + 4.5 × 10^−6^ mol/L H_2_O_2_ + 0.025 mmol/L TMB + 7.5 mg/L FeMOF + 11.6 mg/L AgNPs; b: a + 0.05 nmol/L Apt_GLY_; c: a + 0.1 nmol/L Apt_GLY_; d: a + 1 nmol/L Apt_GLY_; e: a + 5 nmol/L Apt_GLY_; f: a + 0.01 µmol/L Apt_GLY_. (**C**) RRS spectra, a: 0.45 mmol/L Tris-HCl + 4.5 × 10^−6^ mol/L H_2_O_2_ + 0.025 mmol/L TMB + 7.5 mg/L FeMOF + 0.01 µmol/L Apt_GLY_ + 11.6 mg/L AgNPs; b: a + 1 nmol/L GLY; c: a + 2 nmol/L GLY; d: a + 73 nmol/L GLY; e: a + 6 nmol/L GLY; f: a + 8 nmol/L GLY; g: a + 10 nmol/L GLY. (**D**) Fluorescence spectra with excitation wavelength of 300 nm, 2.5 × 10^−4^ mol/L TMB + 5 × 10^−4^ mol/L H_2_O_2_ + 0.45 mmol/L Tris-HCl + FeMOF (a–g: 0, 5 × 10^−4^, 1 × 10^−3^, 1.5 × 10^−3^, 2 × 10^−3^, 3 × 10^−3^, 4 × 10^−3^ g/L FeMOF). (**E**) Fluorescence spectra, 2.5 × 10^−4^ mol/L TMB + 5 × 10^−4^ mol/L H_2_O_2_ + 0.45 mmol/L Tris-HCl + x Apt_GLY_ (0, 0.25, 0.5, 0.75, 1.0, 1.25, 1.5 nmol/L Apt_GLY_). (**F**) Fluorescence spectra, 2.5 × 10^−4^ mol/L TMB + 5 × 10^−4^ mol/L H_2_O_2_ + 0.45 mmol/L Tris-HCl + 1 nmol/L Apt_GLY_ + GLY (0, 1, 2, 3, 6, 8, 10 nmol/L GLY).

**Figure 5 biosensors-12-00920-f005:**
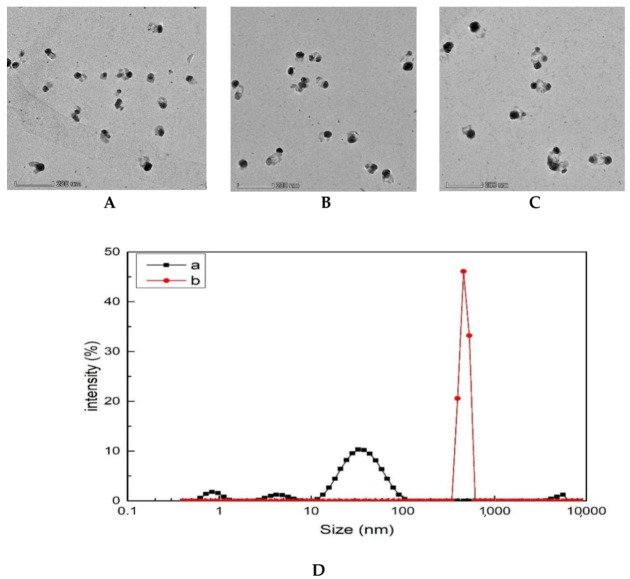
Transmission electron microscopy and laser scattering of FeMOF analytical system. (**A**) 4.0 × 10^−4^ mol/L AgNPs. (**B**) 0.45 mmol/L Tris-HCl + 4.5 × 10^−6^ mol/L H_2_O_2_ + 0.025 mmol/L TMB + 7.5 mg/L FeMOF + 0.01 µmol/L AptGLY + 0.16 µmol/L AgNPs. (**C**) 0.45 mmol/L Tris-HCl + 4.5 × 10^−6^ mol/L H_2_O_2_ + 0.025 mmol/L TMB + 7.5 mg/L FeMOF + 0.01 µmol/L AptGLY + 2.5 ng/L GLY + 0.16 µmol/L AgNPs. (**D**) Laser light scattering; a: blank system; b: FeMOF analysis system.

**Figure 6 biosensors-12-00920-f006:**
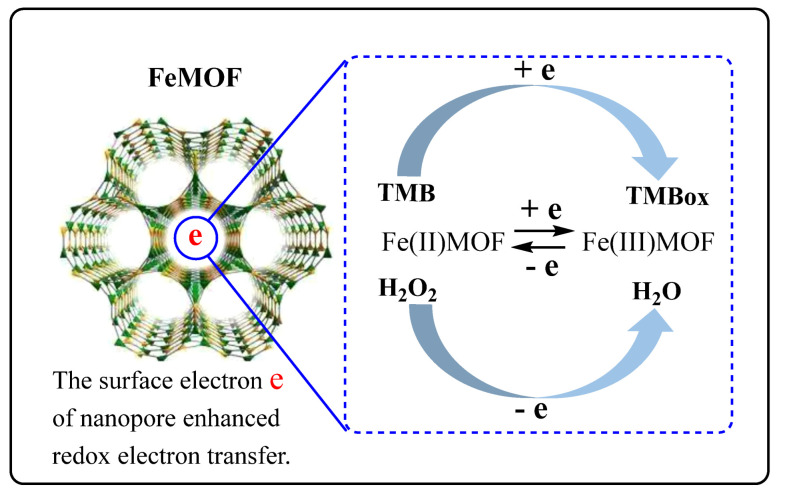
FeMOF nanocatalytic mechanism.

**Figure 7 biosensors-12-00920-f007:**
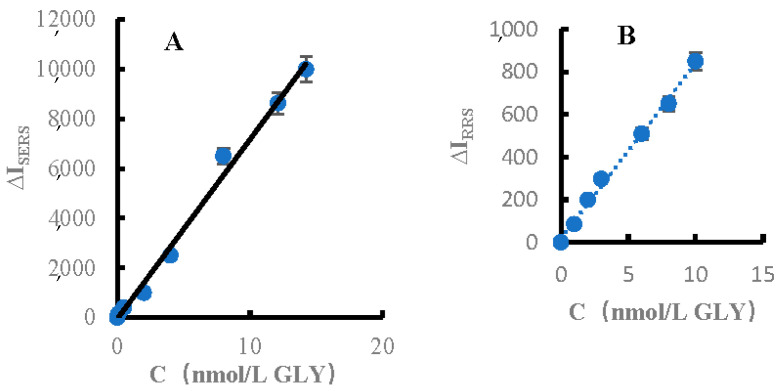
GLY calibration plot of (**A**) SERS, (**B**) RRS and (**C**) FL.

**Table 1 biosensors-12-00920-t001:** Analytical methods for determination of glyphosate.

Method	Linear Range (μg/L)	Detection Limit (μg/L)	Recovery (%)	Sample	Comments	Reference
Colorimetry	1 × 10^3^–2 × 10^4^	1 × 10^3^	100.9	Runoff water	Low sensitivity, but low-cost.	[22]
CE-LIF *	0.169–16.9	0.27	/	River water	Sensitive, but complicated.	[24]
Fluorescence	1 × 10^2^–1.6 × 10^4^	8.75	97.55	Potato	Low sensitivity	[25]
Electrochemistry	0.25–14.0	1.70	/	Type sample	Sensitive and simple.	[26]
LC–MS	0.5–3.03	0.50	91	Human urine	Sensitive, but high-cost.	[30]
UPLC–MS/MS	1–20	0.5	108	Human urine	Sensitive, but high-cost.	[31]
SERS	0.0169–2.37	0.00845	92.1–97.5	Wastewater	Highly sensitive and cheap.	This method

* CE-LIF, capillary electrophoresis laser-induced fluorescence; LC–MS, liquid chromatography–mass spectrometry; UPLC–MS/MS, ultra-performance liquid chromatography–mass spectrometry.

## Data Availability

Not applicable.

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
