# Peer review of "Aptamer Trimode Biosensor for Trace Glyphosate Based on FeMOF Catalytic Oxidation of Tetramethylbenzidine"

_biosensors, 2022, doi:10.3390/bios12110920_

Round 1

Reviewer 1 Report

The investigation shows interesting results with an emphasis to combine many techniques together to study trimode biosensor for trace glyphosate. I recommend the manuscript for its publication. However, the author should revise the manuscript to make it more simple for better understading of the readers. I would like to suggest the following modifications to be done before the final acceptance of the manuscript-

1. Abstract must include need of such biosensor.

2. In abstract, "The addition of silver nanopar- 12 ticles as the substrate produced a strong surface enhanced Raman scattering (SERS) peak at 1607 13 cm-1 and a resonance Rayleigh scattering (RRS) peak at 385 nm.", it is not very clear what about it is mentioned. Please indicate very cleary SERS/RRS of what and what  it signifies ?

3. Introduction: The last paragraph must include the literature regarding such biosensor and special features of Trimode SERS methods.....no such literature has been included. please comments on this and explore this to make it more interesttig for the readers. If possible..provide a schematic of the functioning of such biosensor in general.

4. Role of Ag nanoparticles as plasmonic for SERS should also be emphasized. 

5. Figure 2 has not been well arranged. Arrange properly and explain how the size in figure 2E has been obtained.

6. Many figures has been included in one for example Figure 3 and some plots have been inserted in each sub figures of Figure 3. Please plot seperately and explain or better provide in supplementary.

7. Same with Figure 4. 

8. TEM image should also be provided in the main text along with details in supplementary.

9. Section 5 should be conclusion not discussion.

7. 

Author Response

Point 1: Abstract must include need of such biosensor.

Response: Thanks for your comments! I added “The sensor can be used for the analysis of environmental wastewater, and a new method for detecting glyphosate content in wastewater is proposed. The linear range of the sensor is 0.1 ~ 14 nmol/L, the detection limit is 0.05 nmol/L, the recovery is 92.1 ~ 97.5%, and the relative standard deviation is 3.6 ~ 8.7%.” to the abstract.

Point 2: In abstract, "The addition of silver nanopar- 12 ticles as the substrate produced a strong surface enhanced Raman scattering (SERS) peak at 1607 13 cm-1 and a resonance Rayleigh scattering (RRS) peak at 385 nm.", it is not very clear what about it is mentioned. Please indicate very cleary SERS/RRS of what and what it signifies?

Response: According to this valuable suggestion, we change the sentence "The addition of silver nanopar-12 ticles as the substrate produced a strong surface enhanced Raman scattering (SERS) peak at 1607 13 cm-1 and a resonance Rayleigh scattering (RRS) peak at 385 nm." to “When silver nanoparticles were added, it exhibits strong resonance Rayleigh scattering (RRS) activity and surface en-hanced Raman scattering (SERS) effect.”

Point 3: Introduction: The last paragraph must include the literature regarding such biosensor and special features of Trimode SERS methods.....no such literature has been included. please comments on this and explore this to make it more interesttig for the readers. If possible..provide a schematic of the functioning of such biosensor in general.

Response: According to the good comments, the “The trimode method not only overcome some disadvantages of the single mode methods and has the advantages but also provided three method selection. As far as we know, there is no report on the synthesis of FeMOF with high catalytic activity, nanocatalytic amplification of H2O2-TMB indicator reaction and its coupling with highly selective aptamers to establish three-mode detection.” were added in the last graph of Introduction. There are no reports about the trimode method, and no the schematic of the functioning of such biosensor was added.

Point 4: Role of Ag nanoparticles as plasmonic for SERS should also be emphasized. 

Response: In Section 3.3, the SERS signal enhancement after the addition of Ag nanoparticles is emphasized as follows, “Ag nanoparticle has the properties of surface effect, volume effect, quantum size effect and macro quantum tunneling effect of general metal nanomaterials, and also has the special effect of surface plasmon resonance. When Ag nanoparticle was irradiated by laser, the very small scale of Ag nanoparticle makes the nanosphere cavity array cooperate with surface plasmon resonance and optical coupling, resulting in the enhancement of electromagnetic field. Therefore, the Raman signal of adsorbed molecules on the Ag nanosurface is 106 times or even stronger than that of normal molecules, which makes it one of the best SERS metal substrates.”.

Point 5: Figure 2 has not been well arranged. Arrange properly and explain how the size in figure 2E has been obtained.

Response: The Figure 2 was arranged properly. “The laser scattering size distribution of FeMOF solution was recorded by the nanoparticle size and potential analyzer. The size distribution was from 820 nm to 1200 nm, with an average size of 955 nm (Fig.2E),”. The above was added in the section 3.2.

Point 6: Many figures has been included in one for example Figure 3 and some plots have been inserted in each sub figures of Figure 3. Please plot seperately and explain or better provide in supplementary.

Response: According to this suggestion, the Figure was revised, and the inserted graphs were moved to Figure S1. The content of “The results show that the catalyst concentration was linear to the SERS peak intensity at 1607 cm-1 (Figure S1)” was added in the section 3.3.

Point 7: Same with Figure 4.

Response: Figure 4, like the Figure 3, has been modified.

Point 8: TEM image should also be provided in the main text along with details in supplementary.

Response: The Figure.S1 Transmission electron microscopy and laser scattering of FeMOF analytical system was moved to Figure 5.

Point 9: Section 5 should be conclusion not discussion.

Response: The “Discussion” was changed to “Conclusion”.

Reviewer 2 Report

This manuscript is well written and well organized. In my opinion, it can be accepted after making the following revisions:

1. Authors must review all figures in the manuscript. Overlapping images, illegible letters and numbers due to small size (Fig 2E), missing letters (eg Fig4 D, Fig 4E, Fig 4F). Authors should modify the font size (size like Fig 4D). In all graphs add axis titles. in the main text they must all appear referenced in the same format, for example Fig. 2A or Figure 2A.

2. The authors must add a comparative table with different published works for the determination of glyphosate. The type of analytical technique, linear range, detection limit and analyzed samples must appear, in addition to the reference.

3. In the supplementary information there is no figure or table. Where are the figures S1, S2, S3 and the tables S1, S2?

 4. In Section 3.8 add the glyphosate calibration plot with the spectra, as in Fig 4F. If it is that figure, reference it in the main text, in this section where the linear range is discussed.

Author Response

Point 1: Authors must review all figures in the manuscript. Overlapping images, illegible letters and numbers due to small size (Fig 2E), missing letters (eg Fig4 D, Fig 4E, Fig 4F). Authors should modify the font size (size like Fig 4D). In all graphs add axis titles. in the main text they must all appear referenced in the same format, for example Fig. 2A or Figure 2A.

Response: Thanks for your comments! We have amplified the Fig 2E, revised the overlapping images, supplemented by missing letters, etc.

Point 2: The authors must add a comparative table with different published works for the determination of glyphosate. The type of analytical technique, linear range, detection limit and analyzed samples must appear, in addition to the reference.

Response: In the section of 3.8, we added to the following content, “The Table 1 was listed the reported methods for determination of glyphosate, including the method, detection limit, recovery and analysis sample. This SERS method was sensitive.”.

Point 3: In the supplementary information there is no figure or table. Where are the figures S1, S2, S3 and the tables S1, S2?

Response: The original figures S1, S2, S3 and the tables S1, S2 were uploaded respectively. This revision was uploaded in supplementary information file.

Point 4: In Section 3.8 add the glyphosate calibration plot with the spectra, as in Fig 4F. If it is that figure, reference it in the main text, in this section where the linear range is discussed.

Response: The GLY calibration plot of Figure 7 has been added. The “In the trimode methods, the SERS was most sensitive and the linear range was the widest, the RRS was second and the equipment cost was lower that the Raman meter. In addition, the FM can be finished the RRS/FL dimode detection.” was added in the Section 3.8.

Reviewer 3 Report

The experimental design is ingenious and innovative.

The quality of Figure 3 and Figure 4 needs to be improved.

The format of some references is not standard.

eg. [28] H.B. Shokoth, TY Yang, ZX Li, MH Wu. Preparation of ferrous-doped MIL-53(Fe) photo-Fenton catalyst and its 508 application in dyeing and printing wastewater treatment [J]. Textile Auxiliaries, 2021, 38 (7): 36-40. 509 [29] Qingrun Liu, Chixuan Yao, Jingmin Liu, Shuo Wang, Bing Shao, Kai Yao. An efficient method to enrich, 510 detect and remove bisphenol A based on Fe3O4@MIL-100(Fe)[J]. Microchem. J., 2021, 165: 106168.

Some specific comments on the paper:

1. Figure 3E, The R-squared value of the fitting curve of is 0.8258, which is low. It is recommended to repeat and optimize the experiment.

2. Figure 2A, The absorbance value corresponding to the concentration e is about 1.6, which is not within the range of 0.2~0.8. It is recommended to set FeMOF with different concentration gradients, and reconfirm the relationship between FeMOF concentration and absorbance value.

3. Lack of comparison with the advantages and disadvantages of existing methods, and it is recommended to supplement in the discussion section.

Author Response

Figure 3 and Figure 4 and the reference have been changed.

Response: It was revised.

Point 1: Figure 3E, The R-squared value of the fitting curve of is 0.8258, which is low. It is recommended to repeat and optimize the experiment.

Response: We have paid attention to this problem during the experiment. Under the optimized conditions, the R-squared value of 0.8258 (Figure S1E) was low, due to the un-stability of CuMOF catalyst. It was used to compare FeMOF.

Point 2: Figure 2A, The absorbance value corresponding to the concentration e is about 1.6, which is not within the range of 0.2~0.8. It is recommended to set FeMOF with different concentration gradients, and reconfirm the relationship between FeMOF concentration and absorbance value.

Response: The concentration e has been deleted.

Point 3: Lack of comparison with the advantages and disadvantages of existing methods, and it is recommended to supplement in the discussion section.

Response: The comparison was added in Table 1.

Editor comments

In addition to the review reports, please also revise the section 3.7.2 due to it has a high similarity. After revision, we will check the similarity again.

Response: The section 3.7.2 was revised.

Round 2

Reviewer 1 Report

authors have improved a lot and it can be published now.

Author Response

Thank you for your Suggestions.

Reviewer 2 Report

1. Review figures 3A to 3I, arrows and letters superimposed on the curves.

2. Recheck the citation format of the figures in the main text, there are two types of format (Figure 1. Fig 2E). Select only one format.

Author Response

Point 1: Review figures 3A to 3I, arrows and letters superimposed on the curves.

Response: Thanks for your comments! This is our negligence. We have moved the arrows and letters outside the curve to make the figure clearer.

Point 2: Recheck the citation format of the figures in the main text, there are two types of format (Figure 1. Fig 2E). Select only one format.

Response: The “Fig” was changed to “Figure”.